# The Physiological and Pathological Role of Tissue Nonspecific Alkaline Phosphatase beyond Mineralization

**DOI:** 10.3390/biom11111564

**Published:** 2021-10-21

**Authors:** Saravanan Sekaran, Selvaraj Vimalraj, Lakshmi Thangavelu

**Affiliations:** 1Department of Pharmacology, Saveetha Institute of Medical and Technical Sciences, Saveetha Dental College and Hospitals, Saveetha University, Chennai 600 077, Tamil Nadu, India; lakshmi@saveetha.com; 2Centre for Biotechnology, Anna University, Chennai 600 025, Tamil Nadu, India

**Keywords:** alkaline phosphatase, bone, mineralization, brain, vasculature, calcification

## Abstract

Tissue-nonspecific alkaline phosphatase (TNAP) is a key enzyme responsible for skeletal tissue mineralization. It is involved in the dephosphorylation of various physiological substrates, and has vital physiological functions, including extra-skeletal functions, such as neuronal development, detoxification of lipopolysaccharide (LPS), an anti-inflammatory role, bile pH regulation, and the maintenance of the blood brain barrier (BBB). TNAP is also implicated in ectopic pathological calcification of soft tissues, especially the vasculature. Although it is the crucial enzyme in mineralization of skeletal and dental tissues, it is a logical clinical target to attenuate vascular calcification. Various tools and studies have been developed to inhibit its activity to arrest soft tissue mineralization. However, we should not neglect its other physiological functions prior to therapies targeting TNAP. Therefore, a better understanding into the mechanisms mediated by TNAP is needed for minimizing off targeted effects and aid in the betterment of various pathological scenarios. In this review, we have discussed the mechanism of mineralization and functions of TNAP beyond its primary role of hard tissue mineralization.

## 1. Introduction

Skeletal tissues are extraordinary structures and their biomechanical strength is attributed to the orchestrated process of biomineralization-an intricately controlled event involving the cell driven deposition of hydroxyapatite from ions largely present in body fluids [1]. Physiological mineralization is observed in hard tissues [2,3] and pathological mineralization is widely observed in soft tissues [4,5,6,7,8,9]. Mineralized extracellular matrix is a unique feature of the vertebral system in animals. Bone is a multifaceted organ undergoing remodeling throughout the lifetime by balanced actions of osteoblasts, osteoclasts and osteocytes. Hydroxyapatite (HAP) mineral is hierarchically organized on the type I collagen matrix [10,11]. Osteoblasts are responsible for the laying of organic mineralized matrix. They secrete type 1 collagen which is templated for mineral nucleation and subsequent crystal growth. HAP nucleation results in growth of the crystal following a continuous cross-fibrillar pattern [12,13,14,15]. The hierarchical arrangement of bone structure and model of collagen microfibril along with HAP arrangement is depicted in Figure 1. The process of mineralization is involving direct mechanism mediated by extracellular vesicles (EV’s) released from osteoblasts and an indirect mechanism in which non collagenous proteins that are negatively charged, associate with collagen and direct various mineral precursors for nucleation [16,17]. As a result of the mechanical properties of mineralized tissues, they act as a depot for various minerals that are essential for numerous physiological processes in the body. Alkaline phosphatases (E.C. 3.1.3.1) regulate mineralization in hard tissue under both physiological and pathological conditions. Moreover, they catalyze the dephosphorylation of various physiological and non-physiological substrates [18].

## 2. ALP Isoforms

ALP, first discovered zinc enzyme, belongs to homodimeric metalloenzymes composed of several isoenzymes. Three metal ions such as two Zn^2+^ and one Mg^2+^ are present in each monomer and additionally, five cysteine residues (Cys474, Cys467, Cys183, Cys121 and Cys101) are present in placental alkaline phosphatase (PLAP). The Cys121-Cys183 and Cys467-Cys474 residues form two disulfide bonds, whereas the Cys101 residue remains free. His153, His320, His358, His360, His432, Asp316, Asp357, Asp42, Ser92, Ser155, and Glu311 are among the amino acid residues involved in PLAP’s catalytic activity [20]. Among the two Zn^2+^ ions present on PLAP’s active site, (1) one is directly coordinated with three amino acids (His 320, His 432, and Asp 316), while the fourth coordination is occupied by a water molecule or specific substrate surface. (2) The penta-coordinated second Zn^2+^ ion is coordinated to four amino acid residues: Asp42, Asp357, His358 and Ser92; the fifth coordination site is held by water or a substrate molecule. The hexa-coordinated Mg^2+^ ion consists of three links of three amino acid residues (Asp42, Ser155 and Glu311), and three more links with three water molecules that can be substituted by substrates. In addition, Ca^2+^ ion is required for the correct operation of the enzyme in all mammalian ALPs [21]. Every membrane anchored ALPs isoenzymes are made up of glycoproteins with various gene loci encoding it. (1) tissue nonspecific, (2) intestinal, (3) placental, and (4) germ cell ALP are the four loci that have been discovered so far. In humans, a number of structural, biochemical, and immunologic approaches can be used to distinguish the three enzymes (intestinal, placental and L/B/K) [18].

### 2.1. Liver/Bone/Kidney Alkaline Phosphatase

Tissue non-specific alkaline phosphatase (TNAP) is found in many different organs, but abundant in the skeletal, hepatic and renal tissues. TNAP is a homodimeric protein and in its structure, in addition to one phosphate anion, each monomer is composed of three metallic ions (one Mg^2+^ and two Zn^2+^ cations). Each subunit contains an extended central core β-beach with α-helices, while an extended N-terminal α-helix has a “crown domain”. The crown domain can be characterised as a loose interfacial loop with amino acid residues involved in stabilizing the binding of non-competitive inhibitors to the enzyme [22]. Although it’s likely that their protein moieties are encoded by independent but related genes, differences in electrophoretic mobility and thermal stability between the L/B/K ALPs from different organs are related to variations in post-translational modifications. Additional information is discussed in the section below.

### 2.2. Intestinal Alkaline Phosphatase

The gene responsible for IAP is mapped on the long arm of chromosome 2. IAP is a heat stable isozyme and more active at pH 9.7. Unlike other AP’s the side chain of IAP lacks terminal sialic acid residues [23]. IAP is essential for lipids absorption and is involved in reducing inflammation. IAP suppresses inflammation, particularly by modifying the gut microflora and dephosphorylating LPS [24]. Any alteration in IAP expression and activity increases the vulnerability to inflammation. Inflammatory bowel disease (IBD) is an indication of a condition where the patient’s endogenous IAP production and activity are reduced. Exogenous IAP injection may be a good option for treating IBD and other disorders caused by the disruption of gut microbiota [24,25].

### 2.3. Placental Alkaline Phosphatase

The human placental ALP (PLAP) gene was identified on chromosome 2 and possesses 87% similarity with IAP gene. But, there are differences observed at the carboxyl terminal end of their chains. High concentration of enzyme is seen in placenta and marginal levels are found in blood [26,27]. Neutrophils are responsible for a portion of the serum placental-type ALP. Interestingly, cancer cells can re-express PLAP gene as Regan isoenzyme. In contrast to the other ALP isoenzymes, PLAP is a polymorphic enzyme with up to 18 allelozymes originating from point mutations [25,26,27,28]. As a result of the low catalytic activity in comparison with the other isoenzymes, PLAP is considered to be inefficient due to the neutral pH of placenta. PLAP is highly heat stable and is not active at temperatures below 75 °C [29]. In its homodimeric structure, Asn122 and Asn249 are potential glycosylation sites, and its activity depends on the level of glycosylation. Furthermore, the structure of PLAP consists of two disulfide bridges: the first one stabilises the PLAP orientation, which is situated near the anchor of the Asp481 Glycosylyphosphatidylinositol (GPI) and the second stiffens the anchor of the carbohydrate chain and is situated close the first part of the glycosylation. There are several roles by PLAP, including the transmission of mother-to-fetus immunoglobulins G (IgG). PLAP stimulates DNA synthesis and fibroblast cell proliferation with insulin in the presence of Zn^2+^ and Ca^2+^. It is a synthesiser for a single tissue called syncytiotrophoblast and responsible for transfer of substances such as nutrients and oxygen. PLAP is an essential modulator of foetal growth. PLAP is also a potential marker for various cancers [23].

### 2.4. Germ Cell Alkaline Phosphatase

The germ-cell ALP (GCAP, placental-like ALP) gene has also been localised at chromosome 2, and it is a heat-stable isozyme found at low levels in embryonal tissues, germ cells and some neoplastic tissues [30]. GCAP appears to be limited to the cell membrane of immature germ cells in the testis, and it is attached to the cell membrane by a phosphatidyl-inositol-glycan anchor, as do the other ALP isoenzymes [31]. Cancer cells (or NAGAO isozyme) can re-express it, just like the placental ALP gene [27,31,32]

## 3. TNAP

Tissue-nonspecific alkaline phosphatase (TNAP) is highly expressed on the plasma membrane of osteoblasts, odontoblasts and hypertrophic chondrocytes and also highly concentrated in the extracellular vesicles (EVs) originating from these cells [33,34]. Pyrophosphate (PP_i_) and pyridoxal phosphate (PLP), phosphoethanolamine (PEA) are believed to be the physiological substrates [35,36]. TNAP catalyzes the hydrolysis of PP_i_, an inhibitor of bone mineralization and provides inorganic phosphate ions for hydroxyapatite nucleation and formation [37,38,39]. PLP is essential for the formation of neurotransmitters such as dopamine, histamine, serotonin, taurine and Gamma-aminobutyric acid (GABA) [40]. Pyridoxal administration to Alpl−/− mice prevents epileptic seizures indicating its role in PLP metabolism. PLP entry into the cells is mediated by TNAP which dephosphorylates PLP into pyridoxal and again transformed back into PLP in neurons [41,42]. Unlike PP_i_ and PLP, the confirmed role of phosphoethanolamine as a substrate for TNAP is not known but TNAP deficient patients and knockout mice display elevated levels of phosphoethanolamine [43]. Recent reports also depict that di-phosphoryl lipopolysaccharide (LPS) [44,45], adenosine triphosphate (ATP) [46,47,48] and phosphorylated osteopontin (pOPN) [49,50,51] are TNAP natural substrates.

### 3.1. Gene Structure

Four human ALP isoenzymes are present. TNAP is ubiquitously expressed in bone, liver, kidney, white blood cells and neuronal cells. Human TNAP is encoded by ALPL gene (NCBI Gene ID: 249) which is located on the chromosome 1 short arm (1p36.1-34). The ALPL gene possesses 1.5 kb of coding region, spans more than 50 kb of genomic DNA and contains 12 exons, including 11-proetein coding exons with 2536 transcript length [34,52]. The promoter region of ALPL gene contains a TATA box, Sp1 binding site and retinoic acid responsive element (RARE) which regulates TNAP gene expression [53,54]. The regulation of TNAP expression by retinoic acid is through RARE and active vitamin D is regulated by modifying the stability of TNAP mRNA [55]. In addition to this, phosphates produced by the enzymatic activity of ALP are also known to regulate TNAP expression [56]. The methylation of promoter regions of the ALPL gene is known to be involved in epigenetic regulation [57]. Genes encoding tissue-specific ALPs are present on chromosome 2 long arm and possess a compact gene arrangement [26,58,59].

### 3.2. Protein Structure

TNAP is an ectoenzyme approximately 80kDa, linked to the outer plasma membrane via glycosylphosphatidylinositol (GPI) anchorage [60]. The GPI consists of an ethanolamine phosphate and three glucosamine, mannose, and phosphatidylinositol. It can be sliced by phospholipases present in plasma membranes explaining the circulatory presence of TNAP in various biological fluids [61]. TNAP is synthesized initially as a 66 kDa peptide followed by the addition of O-and N-glycosides in the ER. Among mammalian ALPs 58% of aminoacid residues are highly conserved in human TNAP sequences [62]. Until now, the 3D structure of TNAP is not yet delineated and a simulation-based model of mouse IAP or human PLAP is used to describe the structure (Figure 2). The presence of a catalytic serine residue (S92) is marked in the active site of human PLAP and it also contains two Zn^2+^ binding sites and one Mg^2+^ binding site. A Ca^2+^ binding site is also seen in human and murine TNAP but it does not possess any direct role on the catalytic activity. The crown region is present in mammalian ALPs and known to interact with various extracellular proteins including type 1 collagen [63]. Isoforms of TNAP are distinguished by their distribution pattern on electrophoresis owing to the difference in O-linked sugar chains [60]. Dimerization of two TNAP monomers is mediated by two disulfide bonds that play a prominent role in the severity of hypophosphatasia. TNAP contains five possible N-linked glycosylation sites (N123, 213, 254, 286 and 413) with sugar chains essential for its catalytic activity and types of sugars explains its difference in kinetic and biophysical properties of its isoforms [64].

### 3.3. Mechanism of Mineralization-Promotion and Inhibition

Biomineralization diversifies the mechanical properties of various connective tissues including bone. For instance, mineralization is absent in type 1 collagen of dermis to maintain skin texture, while it is highly mineralized in bone and teeth to form rigid structures [65]. Mineralization of bone follows a two-step process. Initial step involves the formation of HAP inside the EVs and followed by the propagation into the extracellular matrix. The EVs are typically 50-200 nm in diameter which are produced from the plasma membrane of osteoblasts, chondroblasts and odontoblasts [33]. The mechanism behind the release of EVs forms these cells is not yet known. Interestingly, the EVs differ in their membrane composition compared to the originating cell type, marked with the presence of several phospholipids, specifically phosphatidylserine which shows high affinity for calcium ions [66,67,68,69]. EVs are also rich in annexins A5, A2, A6 and calbindin D9k, ALP, carbonic anhydrase, collagen x, type III sodium-phosphate transporter, Phosphatase, Orphan 1 (PHOSPHO1), and nucleotide pyrophosphate phosphodiesterase [70,71,72,73]. Calcium binding proteins, phospholipids, and bone sialoprotein (BSP) mediate calcium accumulation in EVs [74]. Incorporation of calcium in EVs is facilitated by the formation of calcium channels by membrane bound annexins. The role of various annexins and matrix vesicles in bone mineralization is discussed in depth and more insights can be acquired by reading Ansari et al. [75]. Next, phosphates are provided by type III Na/P_i_ cotransporter present on both EVs membrane and cell membrane [76,77]. In addition to this, PHOSPHO1, a cytosolic phosphatase also contributes to phosphate levels by catalyzing the hydrolysis of phosphocholine and phosphoethanolamine [78,79]. When the accumulation of both calcium and phosphate exceeds solubility point for calcium phosphate it leads to the deposition as hydroxyapatite inside the EVs. In the next step of mineralization, hydroxyapatite crystals penetrate EVs membrane to reach the extracellular space and undergo crystal elongation. Elongation is highly dependent on extracellular concentrations of calcium and phosphate ions outside the EVs [73,75]. Sufficient levels of calcium and P_i_ support the formation of new apatite crystals, which propagate in clusters around the EVs and fill in between collagen fibrils in the ECM in a cross-fibrillar pattern [19]. The mechanism of mineralization is diagrammatically represented in Figure 3.

Propagation around clusters is mediated cooperatively by EVs and adjacent collagen molecules. Since mineralization is central to bone, our body prevents ectopic biomineralization by various ingenious inhibitors. In bone, to eliminate the inhibitors of biomineralization especially PP_i_, osteoblasts express TNAP to annul its inhibitory effect via hydrolysis and generate phosphate ions promoting mineralization. TNAP not only provides phosphate ions for mineralization it also promotes the formation of HAP nanocrystals in the collagen matrix of bone by destabilizing the amorphous phase of the mineral. Also, local inhibitors of biomineralization attenuates pathological mineralization in soft tissues such as cardiac arteries, valves and hard/soft tissue interfaces such as tendon-bone attachments, ligament-bone attachments and cranial sutures. It is noteworthy to mention that, the inorganic phosphate (P_i_) to pyrophosphate (PP_i_) ratio is crucial for mineralization as PP_i_ inhibits hydroxyapatite formation [80]. ANKH, a human homolog of ank gene product, is essential for the extracellular transportation of cytosolic PP_i_ [81,82]. An increase in PP_i_/P_i_ ratio will inhibit spontaneous precipitation of HAP. The PP_i_ concentration is rightly maintained in forming bone by osteoblasts and acts as a repository of compartmentalized phosphate till it is hydrolyzed [80]. Extracellular polyphosphate (polyP_i_) also participates in phosphates transport and compartmentalization. Poly-P_i_ granules chelate calcium ions to form neutrally charged amorphous complexes [83,84]. Fetuin A is a systemic inhibitor of mineralization which prevents growth of nascent crystal nuclei in blood and facilitates its recycling by macrophages [85,86,87,88]. One fetuin is known to sequester 54–72 phosphate ions and 90–120 calcium ions. Fetuin A possesses strong affinity to bone and constitutes 25% of the non-collagenous proteins in bone [89]. It binds with both Ca and PO_4_^2−^ ions to form calciprotein particles about 30–150 nm in size [90]. TNAP, expressed highly in osteoblasts, is very essential for the second step of mineralization by decreasing the levels of PP_i_ and providing P_i_ for hydroxyapatite formation. Therefore, in bone, mineralization is a well-coordinated, cell mediated dynamic process allowing the exchange of phosphate and calcium ions. In soft tissues, mineralization is generally inhibited by high PP_i_/P_i_ ratio. The polyP_i_ granules are also known to contain alkaline phosphatases and when activated, hydroxyapatite crystals nucleate inside the granules displacing protein components to the surface forming a crystalline core surrounded by an amorphous shell [84]. Skeletal tissue biomineralization is also promoted by bone sialoprotein (BSP), i.e., a calcium-binding small integrin-binding ligand N-linked glycoprotein (SIBLING) aids in nucleation of hydroxyapatite mineral [91]. It is only present in the ECM of mineralized tissues such as bone and teeth. Another promoter of ECM mineralization, Dentin matrix acidic phosphoprotein1 (DMP1) is involved in the stabilization of disordered mineral precursors and directs them to collagen fibrils [92,93]. DMP1 binds with mineral forming mineral-protein complexes facilitating electrostatic interaction with collagen fibrils during mineralization. Patients with DMP1 gene deletions or mutations are characterized by osteomalacia with autosomal recessive hypophosphatemic rickets [94]. OPN and matrix Gla protein (MGP) are mineralization inhibiting proteins which regulate mineral growth in bone and dentition to adjust mineralization or involve complete inhibition of mineralization [95,96]. Nascent mineralization foci are associated with osteopontin (OPN) and found widely in the bone matrix. For instance, OPN deficient mice exhibit a hypermineralized skeleton postnatally starting at 12 weeks [97]. OPN plays a crucial role as interfacial protein in adjoining new bone to old bone and also to implant surfaces following bone graft surgeries. It is actively found in sites where mineralization should be abolished abruptly such as periodontal ligament, and entheses [98]. OPN knock-out mice failed to curb crystal growth with increased size of mineral crystals but do not show any visible skeletal changes [99].

Despite the precise inhibition of mineralization at soft-tissue sites, arteries and articular cartilage are susceptible for ectopic mineralization which leads to detrimental effects. Therefore, mineralization inhibition requires strict regulation involving additional elements. Vascular smooth muscle cells and chondrocytes orchestrate this inhibition by secreting Matrix Gla protein (MGP) into their extracellular matrix (ECM) [100]. Singleton Merten syndrome and Keutel syndrome associated with defects in MGP gene results in rupture of the aorta and premature fusion of growth plates in the long bone due to pathological mineralization [101]. Other mineralization inhibitors such as aspartic acid-rich motif (ASARM) and matrix extracellular phosphoglycoprotein (MEPE) also fine tune mineralization [102,103]. MEPE knock out mice show increased bone mass and trabecular density but abnormalities in cancellous bone [104]. MEPE along with dentin matrix protein (DMP1) and Phosphate-regulating neutral endopeptidase (PHEX) regulates mineralization, phosphate levels and turnover of bone by affecting fibroblast growth factor (FGF)-23 expression [105]. Imbalance in mineralization promotion and inhibition at required areas in the body leads to various pathologies. The following sections, we will discuss the pivotal role of TNAP in various organs with respect to mineralization.

### 3.4. Hypophosphatasia-Physiological Substrates of TNAP

In humans, endochondral ossification and intramembranous ossification is dependent on TNAP activity. A common manifestation of loss of TNAP activity or mutations to ALPL gene causes a systemic bone disease called hypophosphatasia (HPP) resulting in hypomineralization of hard skeletal tissues including bone and teeth [106,107]. Several missense mutations are reported in human ALPL gene corresponding to aminoacid changes in the protein [106,107,108]. At the time of writing this review, 409 mutations in TNAP have been reported owing to its clinical heterogeneity. The mutant form of TNAP possesses variable kinetic properties for PLP and PPi [109]. The clinical expression of this disease is highly variable affecting patients of all ages. It is a multisystemic disease affecting various tissues such as bone, kidney, muscle, GI tract, lung and central nervous system (CNS). Milder forms of the disease include deformities of extremities, early deciduous teeth loss and other dental manifestations. In severe cases of HPP, prenatal death of foetuses is seen due to complete absence of minerals in their skeleton [106,107,108]. Epileptic seizures, craniosynostosis, respiratory failure and death are also associated in extreme forms of HPP. As far as treatment regime is concerned, there are no curative treatment options available for HPP and it is treated symptomatically requiring a multidisciplinary team and approach [110]. Enzyme replacement therapy (ERT) includes Human recombinant enzyme Asfotase alfa which is approved for patients with pediatric onset of HPP to treat various skeletal and respiratory manifestations (NCT02797821). ERT improves growth, motor function, agility, strength and reduced pain in children with HPP [111]. Patients with a severe life-threatening form of HPP also showed substantial improvement in bone mineralization and survival. In humans and mice models of HPP, the TNAP-deficient EVs’ extracellular growth of HA crystals is blocked by excessive extracellular accumulation of PP_i_ [112]. Combined ablation of PHOSPHO1 and TNAP results in the absence of HA crystals within EVs, absence of skeletal mineralization and embryonic deadliness [113]. Between TNAP and PHOSPHO1 exists a cross talk in the mineralization initiation [114]. PHOSPHO1 is essential for P_i_ generation within MVs which is necessary for the initiation of HA formation inside the vesicle. TNAP is a crucial PPiase essential for the extracellular growth of HA by supplying P_i_ via PP_i_ hydrolysis [112]. Ecto-Nucleotide Pyrophosphatase/ Phosphodiesterase-1 (NPP1), a cell surface enzyme on EVs is a potent ATPase which produces PPi and acts as a phosphatase in the absence of TNAP [115]. NPP1 may modify HPP phenotype in experimental models. For instance, in PHOSPHO1 deficient mice, increase PP_i_ levels was detected in plasma caused by reciprocal decrease in TNAP activity and elevated NPP1 expression. Interestingly, pOPN circulatory levels were increased, an inhibitor of mineralization which furthers the risk of fracturing in HPP [50]. The convergent step involving influx of P_i_ produced by NPP1 and TNAP and production of P_i_ by PHOSPHO1 inside the EVs suggests the combined interactions between TNAP, NPP1 and PHOSPHO1 during EVs mediated calcification [50].

By investigating HPP patients, PLP, which is a circulating form of vitamin B6, was found to be another natural substrate of TNAP. PLP is coenzyme essential for various neurotransmitter’s synthesis [116] and its entry into neuronal cells is promoted by removal of Pi mediated by TNAP [22]. Increased plasma PLP levels were seen in HPP patients with TNAP deficiency. In the severe form of HPP, infants display vitamin B6-dependent seizures resulting from low circulatory levels of PL and inefficient incorporation into CNS [117,118]. The importance of TNAP in CNS is shown by its presence in developing murine neural tube and areas of the mature brain [119,120]. Increased amounts of less developed cortical synapses, hypomyelination and spinal nerves thinning are seen in TNAP KO mice [41,42,121]. Ingestion of PL temporarily suppressed epilepsy in TNAP KO mice. This evidence suggests the role of TNAP in both developing and mature CNS.

Another biological substrate of TNAP is PEA, whose level was elevated in urine and blood of HPP patients [122]. PEA is a component of glycosylphosphatidylinositol link for proteins like TNAP. TNAP is essential for the extracellular hydrolysis of PLP to PL and transport into cells for PLP formation, a vital cofactor in many reactions. Hepatic O-phosphorylethanolamine phospholyase (PEA-P-lyase) enzyme is reported to be involved in PEA hydrolysis using PLP as cofactor [123]. In HPP patients, insufficiency in the intracellular levels of PLP in hepatic cells could be the reason for elevated PEA accumulation and its rise in blood and urine levels. Despite this, the exact mechanism is yet to be known behind this PEA accumulation in HPP patients.

Yet another pathogenicity in HPP is increased plasma OPN (encoded by Spp1) levels suggesting that it may be another potential substrate of TNAP. Phosphorylation of OPN inhibits mineral deposition [124]. However, the exact role of OPN is not completely understood. OPN is known to anchor osteoclasts to HA minerals by poly-aspartate sequences and bind CD44 and αvβ3 integrin with RGD, which mediates cell migration and cell signaling [125]. Phosphorylated form of OPN inhibits mineralization in vascular smooth muscle cells [126]. High levels of phosphorylated OPN and extracellular PPi levels were observed in Alpl KO mice. Double knockout of Alpl^−/−^ and Spp1^−/−^ in mice models partially improved hypomineralization compared to Alp^−/−^ KO mice [127]. Therefore, TNAP loss leads to accumulation of phosphorylated OPN which in turn results in impaired bone mineralization in murine HPP. Similarly, PHOSPHO1 deficient mice models exhibited an increase in levels of both PPi and phosphorylated OPN. Ablation of Spp1 reverts the skeletal deficiency in Phospho1 deficient mice [50]. Altogether, it is clear that TNAP exhibits a wide range of substrates and whose levels are disrupted in TNAP dysfunction.

### 3.5. Role of TNAP as An Anti-Inflammatory Enzyme

TNAP is available as a soluble isoform in blood primarily originating from bone and liver tissue. Bone is the main source of TNAP during skeletal growth and it slowly progresses down with the aging process, thus the liver becomes the major source of TNAP in blood. It is very challenging to distinguish between bone and liver TNAPs in blood as bone TNAP exhibits 18% cross reactivity with liver TNAP [128,129]. Circulating TNAP exert an anti-inflammatory role by contributing to the generation of adenosine from AMP, detoxification by LPS dephosphorylation and regulating postprandial endotoxaemia [130,131]. In the HPP model with TNAP deficiency, bone marrow edema [132], osteomyelitis [133], tendinitis [134] and increased predisposition to periodontitis is widely seen among both children and adults [135]. TNAP through its ectonucleotidase activity exerts a major role in balancing pro-inflammatory ATP levels and anti-inflammatory role through adenosine, a breakdown product of ATP [136]. Owing to this, it has gained much attention leading to various studies investigating several agonists and antagonists. Impairment in TNAP ectophosphatase activity leads to PPi accumulation initiating the formation of calcium pyrophosphate dihydrate crystals [137]. These crystals which accumulate in tissue and joints of HPP patients trigger necroinflammation and NLRP3 inflammasome activation [138]. OPN is a natural substrate of TNAP is thought to be involved in mineralization. However, it is a potent proinflammatory protein and status of pro- and anti-inflammatory properties are attributed to the status of its phosphorylation and dephosphorylation mediated by TNAP. Very recently, recombinant OPN was found to mediate anti-inflammatory cascades in microglia of the brain by inhibition of NLRP3 inflammasome [139]. OPN also regulates inflammatory processes in kidney and liver. However, in-depth investigation dissecting the role of TNAP mediated OPN dephosphorylation in balancing pro- and anti-inflammatory effects is necessary. The ectophosphatase activity of TNAP is also associated with the dephosphorylation of TLR ligands, such as double-stranded RNA; mimic poly-inosine:cytosine, a TLR3 agonist; and microbial LPS, a TLR4 ligand that mitigates the activation of inflammasome and the secretion of cytokines in sepsis [131]. TNAP is also known to modulate T-cell function in a preclinical model of intestinal colitis [140]. TNAP balances P1 and P2 receptor mediated signaling in modulating the level of the inflammatory process. TNAP present in the neutrophil cell membrane hydrolyze AMP, LPS and PLP and controls autocrine effects of adenosine on neutrophil migration, survival and IL-1α secretion [136,141,142]. TNAP in the membrane of endothelial cells may be involved in LPS dephosphorylation [131]. In 7-day old Alpl^+/−^ mice bone metaphysis, the levels of IL-1α and IL-6 were increased and IL-10 anti-inflammatory levels were decreased compared with Alpl^+/+^ mice. In hypertrophic chondrocytes, TNAP inhibition had no effect on ATP and adenosine associated changes including the modulation of autocrine pro-inflammatory effects. In neutrophils, inhibition of TNAP worsened ATP induced secretion of IL-1α and reduced cell survival [136]. AP infusion in several phase 2 clinical trials has shown improvement in the outcome of various inflammatory diseases such as sepsis, inflammatory bowel disease and ischemia/reperfusion [143]. Overall, the role of TNAP in modulating inflammation and further investigations are needed to decipher the exact mechanisms behind its mode of action.

### 3.6. Role of TNAP in Central Nervous System

TNAP is also expressed by neuronal and endothelial cells where it plays a crucial role in brain development. Interestingly, TNAP is very active in the arterial part but not at the venal part of the microvasculature [144,145]. It can be detected in human brain blood vessels right from gestational ages and this distribution of TNAP hints its involvement in active transport of molecules across the blood brain barrier [146,147]. It is very active in both luminal as well as abluminal sides of the endothelial cell membrane mediating transport across these cells. The possible role of TNAP includes PLP dephosphorylation and pyridoxal transport across epithelial cells of capillary. Epileptic seizures, decreased GABA production and elevated levels of PLP in blood of Alpl-deficient mice with HPP dictates the vital role of TNAP in the nervous system. Apart from endothelial TNAP, its presence in neurons is very important in normal brain development as it is widely seen in both white and grey matter during developmental stages of the brain [148]. Surprisingly, elevated TNAP activity is also observed in adult neurogenesis. Delayed myelination and spinal abnormalities are seen in Alpl-deficient mice. MRI investigations of HPP infants shows hypodensity of white matter, multicystic encephalopathy, dilated ventricles, and parenchymal lesions [149,150]. Dephosphorylation and interaction with ECM proteins including collagen and laminin is also contributed by TNAP in brain development [151]. Post mortem of the brain from patients with Alzheimer’s disease (AD) showed increased TNAP protein levels and its activity in temporal gyrus and hippocampus, regions targeted by tau protein accumulation [152,153]. TNAP may exhibit dual effects in AD. First, it ameliorates neuroinflammation by adenosine synthesis through ATP dephosphorylation and maintains a normal functional BBB [154]. In contrast, it participates in dephosphorylation of hyperphorylated extracellular Tau protein which interacts with muscarinic receptors disrupting calcium homeostasis which leads to neuronal death [153]. TNAP inhibition may mediate brain damage induced by ischemic stroke events [155]. Therefore, it is regarded as an important biomarker and mediates events following ischemic reperfusion injuries. Patients with acute ischemic stroke show an elevated level of TNAP in serum which is associated with stroke recurrence and death [156,157,158,159]. On the other hand, TNAP may also attenuate neuroinflammation after stroke by dephosphorylation of ATP which is released in large amounts following cell necrosis [160]. Thus, TNAP blockade may lead to worsening of brain damage and contribute to neuroinflammation and therefore, interventions on TNAP inhibition require a thorough investigation.

### 3.7. Hepatic Role of TNAP

Although the expression of TNAP is known, its function remains obscure in the kidney and liver. In hepatocytes, TNAP is located at the canalicular membrane and in the apical area in the cytoplasm, in bile duct of epithelial cells [161,162]. It is involved in bile excretion and participates in bile pH regulation by dephosphorylating ATP at the surface of cholangiocytes [163]. Circulatory levels of liver TNAP is largely seen in cholangitis suggesting its role in bile excretion. In addition to this, it may involve detoxification and biliary excretion of LPS [164]. During systemic inflammation, in adults, TNAP from the liver is largely released into the blood when bone formation is compromised and bone TNAP levels are reduced in blood. However, the exact function of hepatic TNAP is still unknown.

### 3.8. Renal Role of TNAP

In the kidneys, proximal renal tubules’ brush border expresses TNAP and is involved in LPS detoxification by dephosphorylation [164,165]. Renal tissue express TNAP, an activator of mineralization. One would expect mineralization in renal tissue due to the presence of TNAP but interestingly, urine mineralization is prevented by PP_i_. The PPi production site is at the distal nephron; there is no TNAP expression there, which explains the inability of TNAP in the renal system to trigger urinary tract calcification [166]. TNAP inhibitors blocked norepinephrine dependent renovascular and blood pressure responses indicating the importance of TNAP in the kidney. The detrimental role of TNAP is seen in chronic kidney disease (CKD) where its activation leads to arterial media calcification resulting in stiffening of the blood vessels [167,168]. In CKD, calcification inhibitors such as MGP and PP_i_ are decreased which results in ectopic mineralization of the artery wall [169]. The precise balance between PP_i_/P_i_ is perturbed leads to hydroxyapatite mineralization. Inhibition of TNAP activity leads to an improvement in CKD pathologies. Additional information on vascular calcification is discussed below.

### 3.9. TNAP in Vascular Calcification

Cardiovascular calcification (CVC) is an important risk factor for morbidity and mortality in patients of all ethnicities and increases with age [170]. Age related calcification includes aortic valve calcification and atherosclerotic plaque calcification and in chronic kidney disease and type 2 diabetes, tunica media calcification is largely seen to be associated with increased mortality risk [171]. Cardiovascular mortality risk is positively linked to calcium levels [172]. Until recently, TNAP was found to be a central player in CVC and opened new avenues for treating and managing the disease. All CVC types mimic either endochondral or intramembranous calcification and both have been seen in the arterial tunica media of both rodents and humans with CKD and diabetes (Figure 4). Both physiological and pathological calcification share the same molecular events leading to ECM mineralization.

Calcifying vascular specimens shows the presence of osteoblasts, osteoclasts and chondrocytes which were derived from stem cells or by transdifferentiation of vascular smooth muscle cells [173,174,175,176]. Local TNAP activation triggers massive vascular calcification in arteries owing to its overexpression in vascular smooth muscle cells and endothelial cells [177,178]. Likewise, circulating TNAP is also associated with increased mortality in patients with CKD where the serum TNAP is found to be elevated. In patients undergoing hemodialysis, plasma pyrophosphate levels are reduced following dialysis and it is associated with aortic wall vascular calcifications [179,180]. Involvement of the TNAP/PP_i_ system is interesting and is worth investigating to subdue and inhibit VC. Both TNAP and NPP1 enzymes are present on the cell membrane of calcifying cells to precisely regulate levels of PP_i_, inhibitor of mineralization by blocking the expansion of nascent hydroxyapatite crystals. Routine administration of exogenous pyrophosphate prevented vascular calcification (VC) in mice and rat models [181,182]. In mice models with ank mutation or NPP1-/- exhibit vascular changes through the alteration of cartilage specific genes. In humans, NPP1 deficiency causes idiopathic infantile arterial calcification which affects children resulting in medial calcification of arteries [183,184]. Early microcalfications is targeted by bisphosphonates (non-hydrolysable PP_i_ analogues) preventing vessel wall mineralization. TNAP activity is more prominently observed in uremic rats and mouse models of medial calcification [185]. The dual functions of ALP well known to release P_i_ by phosphomonoesters hydrolysis and act as pyrophosphatases in bone. However, its function outside bone is uncertain. In HPP patients, with elevated TNAP activity there is no clear evidence of vascular calcification indicating that only membrane bound TNAP and other cofactors are required to induce VC. Finally, transdifferentiation of SMCs to chondrocytes via BMP-2 activation and calcium deposition is stimulated by TNAP which explains the role in VC [186]. Overexpression of TNAP induces medial vascular calcification in an ex vivo model of rat aortic rings [9,187]. Although pharmacological inhibition of ALP and PHOSPHO1 also suppressed vascular smooth muscle cell calcification, it induced loss of skeletal mineralization [188,189]. Despite various reports, it is very hard to conclude the involvement of amplified TNAP levels in vascular calcification. Greater understanding of the mechanisms involving TNAP in vascular calcification is needed to better study the clinical consequences and develop therapeutic agents.

## 4. Pharmacological Inhibition of TNAP Needs Careful Consideration

Stable PP_i_/Phosphate ratio is crucial in maintaining physiological bone mineralization and to protect soft tissue ectopic calcification. Increased TNAP expression leads to increased degradation of PP_i_ into phosphate ions thereby generating a procalcifying microenvironment in the vessel wall which leads to calcification in arteries [190]. Until recently, TNAP inhibitor levamisole was used to prevent CVC but it had TNAP independent effects on voltage gated sodium channels [191]. MLS-0038949 (arylsulfonamide 2,5-dimethoxy-N-(quinolin-3-yl) benzenesulfonamide) is a selective TNAP inhibitor and do not inhibit IAP but had very modest pharmacokinetic properties and only tested in vitro on cultured VSMCs [189]. Its ability to differentially inhibit bone and liver TNAP (differing only in the glycosylation residues) has not yet been explored. JL Millan and his team developed a potent selective TNAP inhibitor SBI-425, inhibited plasma TNAP activity and decreased arterial calcification [192]. Single dose of SBI-425 at 10mg/kg/day administered orally or through an intravenous route produced promising results in the TNAP over expressing mice. SBI-425 at 10 or 30mg/kg/day also exerted a positive role in preventing VC in mice models, impairing PP_i_ generation and preventing aorta calcification in CKD mice fed with phosphorous and an adenine rich diet [168]. Oral administration of SBI-425 decreased accumulation of coronary calcium and LV hypertrophy in wicked high cholesterol mouse models [178]. An alternative approach to lessen TNAP activity is by targeting it at the protein level. In this aspect, an orally available bromodomain and extraterminal protein inhibitor namely apabetalone used in CVD treatment was found to be a potent TNAP inhibitor at mRNA level [193]. It reduced ALPL mRNA levels, TNAP protein, and the activity of human hepatocytes and VSMCs, leading to reduced VC. Apabetalone also diminished TNAP levels in circulation in CVD and CKD patients [194]. However, it had no cardiovascular beneficial effect in patients with type 2 diabetes, acute coronary syndrome and low HDL levels. TNAP level is also elevated in patients with MI both in serum and hearts. Rat MI models also indicated the elevation of TNAP and subsequent administration of tetramisole, TNAP inhibitor improved cardiac function and reduced fibrosis post MI [195]. TNAP inhibition in neonatal rat cardiac fibroblasts attenuated the expression of collagen related genes through TGF-1/Smad signaling suppression and p53 and p-AMPK upregulation [196]. In a rat model, warfarin was used to induce calcification in the aorta and other peripheral arteries, suggesting a protective role of SBI-425 [197]. Oral administration of SBI-425 decreased calcium content in aorta and peripheral arteries with discernible reduction in calcified areas. However, it decreased the rate of bone formation, mineral apposition and lengthened osteoid maturation time which opens up the platform for debate regarding its clinical translation. Owing to the fact that TNAP is expressed ubiquitously in the body with various bodily functions (Figure 5), its pharmacological inhibition needs controlled evaluation in vivo prior to its clinical translation.

## 5. Conclusions

TNAP, an ubiquitously expressed enzyme, is ancient yet an active enzyme with multifaceted roles in the body. It is involved in various physiological and pathological conditions owing to its enzymatic activities utilizing varieties of substrates. The well-established function of maintaining PPi/Pi ratio is crucial. Considering the dual role of TNAP, it is a double-edged sword with both positive and negative effects based on the working microenvironment. Activation at unintended sites such as soft tissues and inactivation or loss of function at intended sites mainly in bone leads to detrimental effects. Although there is an unbalanced view on the importance of TNAP in the past, there is a pressing need to perform various studies to better study and understand associated functions and design therapeutic regimens.

## Figures and Tables

**Figure 1 biomolecules-11-01564-f001:**
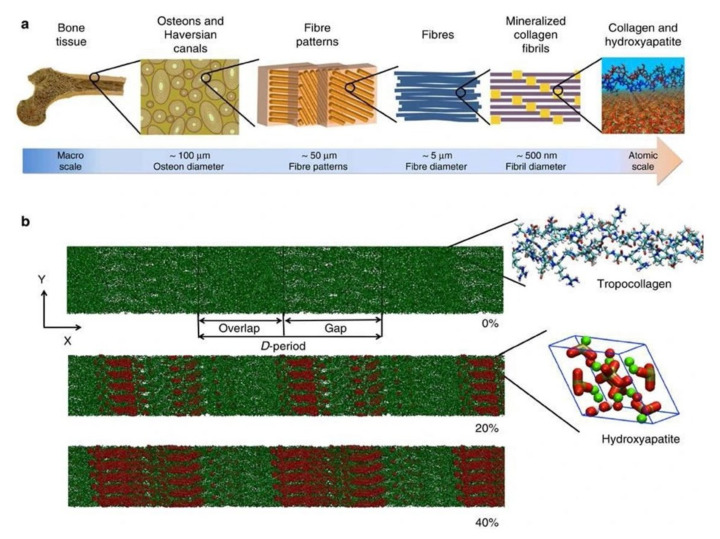
(**a**) Hierarchical structure of bone depicting the arrangement from macro to nanoscale. (**b**) Collagen microfibril model indicating different levels of mineralization from 0% to 40% indicating the alignment of hydroxyapatite crystals. The c-axis of HAP aligns with the collagen fibril axis. Green color indicates Ca atoms, red and white denote OH groups and phosphate groups are visualized in the tetrahedron structure. The image was taken from Arun et al. [19].

**Figure 2 biomolecules-11-01564-f002:**
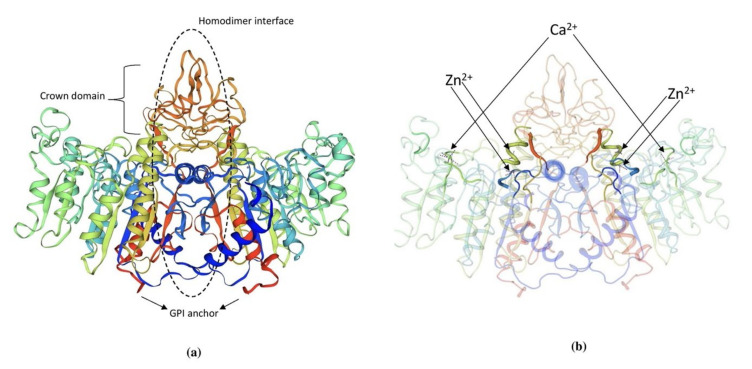
(**a**) 3D structure of human PLAP indicating crown domain, GPI anchor and the homodimer interface regions. (**b**) The presence of zinc and calcium ions is clearly indicated in the PLAP structure. The structure was exported from https://swissmodel.expasy.org/, accessed on 1 August 2021. (P05186 (PPBT_HUMAN)).

**Figure 3 biomolecules-11-01564-f003:**
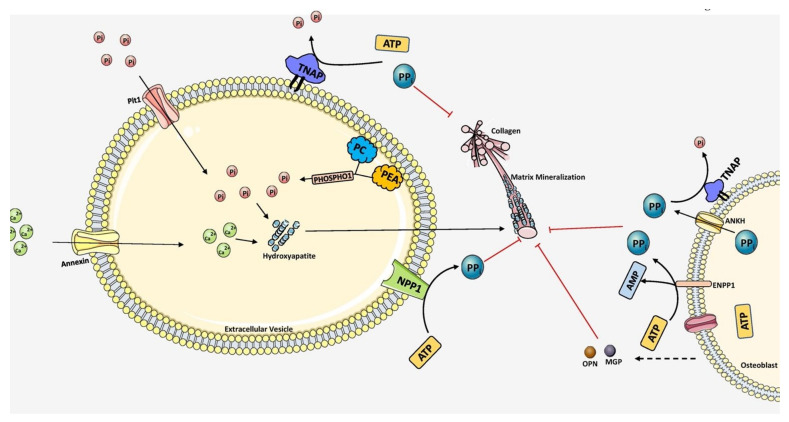
Overall scheme of matrix mineralization mediated by extracellular vesicles (EV) released from osteoblasts. The growth of hydroxyapatite inside EVs is mediated by the PHOSPHO1 which supplies P_i_ by hydrolyzing PC and PEA in the plasma membrane. P_i_t1 transporter also pumps P_i_ into the EVs. Calcium is transported inside by Annexin (A1, A2, A4, A5, A6, A7) channels. Together, hydroxyapatite crystals are formed inside the EV. It then penetrates through the vesicle membrane and elongates extracellularly utilizing P_i_ and Ca^2+^ ions. PP_i_ inhibits mineralization which is produced by ATP hydrolysis mediated by NPP1. TNAP hydrolyzes PPi into P_i_, essential for hydroxyapatite growth. Extracellular PP_i_ is also provided by ANKH and ENPP1 situated in the plasma membrane of osteoblasts. Mineralization is regulated tightly by maintaining the balance between PP_i_ and P_i_ ratio. The components in this figure were modified from Servier Medical Art, licensed under a Creative Common Attribution 3.0 Generic License. (http://smart.servier.com, accessed on 1 August 2021).

**Figure 4 biomolecules-11-01564-f004:**
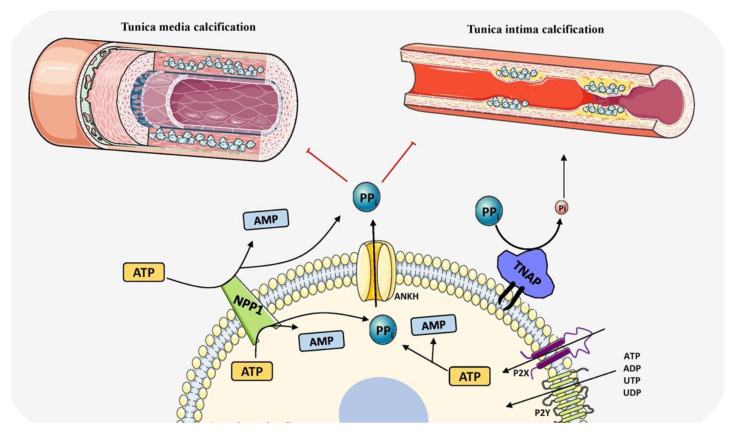
Schematic representation of the role of TNAP in vascular calcification by determining PP_i_/P_i_ ratio and balancing induction and inhibition of mineralization. The components in this figure were modified from Servier Medical Art, licensed under a Creative Common Attribution 3.0 Generic License (http://smart.servier.com, accessed on 1 August 2021).

**Figure 5 biomolecules-11-01564-f005:**
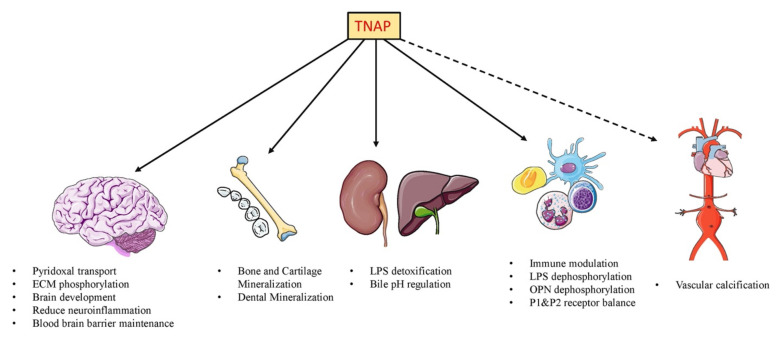
Diagrammatic representation of the central role of TNAP in our body exhibiting mineralization and non-mineralization functions. Dotted arrow indicates its pathological role. The components in this figure were modified from Servier Medical Art, licensed under a Creative Common Attribution 3.0 Generic License (http://smart.servier.com, accessed on 1 August 2021).

## Data Availability

Not applicable.

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
