# Peer review of "The Physiological and Pathological Role of Tissue Nonspecific Alkaline Phosphatase beyond Mineralization"

_biomolecules, 2021, doi:10.3390/biom11111564_

Round 1
Reviewer 1 Report
This manuscript is a very interesting and adequate review about multifaceted roles of TNAP, but some limitations of it are visible:
Not enough references about annexins are cited, only [77] is described in the manuscript (line 195).
At Fig.3 only AnxA5 is indicated and what about others like AnxA6?
Ca2+ and PPi or Pi should be written as superscript and subscript in Fig. 3 and Fig. 4 legends and in all manuscript.
TNAP indication is ommitted at Fig. 3 and Fig. 4.
In vitro (line 514) and in vivo (line 543) words should be written in Italic style.
At Fig. 5 it is worth to add bone, cartilage and dental mineralization.
Author Response
Comment: Not enough references about annexins are cited, only [77] is described in the manuscript (line 195).
Response: We thank the reviewer for his valuable comment. The manuscript primarily focussed on the TNAP role and therefore the role of annexins was limited in the manuscript. However, we have included a recent manuscript that will provide more insights to the readers regarding MVs and Annexin on matrix mineralization.
Comment: At Fig.3 only AnxA5 is indicated and what about others like AnxA6?
Response: We appreciate the reviewer for the suggestion. The figure legend is now modified and includes other annexins also.
Comment: Ca2+ and PPi or Pi should be written as superscript and subscript in Fig. 3 and Fig. 4 legends and in all manuscript.
Response: We apologize for these mistakes. It is now corrected in the revised manuscript
Comment: TNAP indication is ommitted at Fig. 3 and Fig. 4.
Response: We thank the reviewer for rightly pointing out the omission. We have now corrected the revised manuscript
Comment: In vitro (line 514) and in vivo (line 543) words should be written in Italic style.
Response: We apologize for the mistake. It is now corrected in the revised manuscript.
Comment: At Fig. 5 it is worth to add bone, cartilage and dental mineralization.
Response: We thank the reviewer for the suggestion. As suggested it is now included in the manuscript.
Reviewer 2 Report
Please revise your manuscript to correct all of the grammatical errors.
Line 65 – Should be Mg2+. All charges should be superscripts.
Line 141 – “Table X”- what or where is Table X?
Line 237 – “PO4” should be PO42-
Line 246 - “promoted by bone sialoprotein (BSP), a calcium is a calcium-binding small integrin-binding ligand…”. Please rephrase this to clarify your meaning.
Line 404- “TNAP in inevitable…”. I’m not sure what you are trying to say here but “inevitable” doesn’t seem to be the correct word?
Line 488 – “NNP1-/-“, should this be NPP1-/-?
Line 553 – “Pi/PPi ratio”, you have previously referred to this in your manuscript as the PPi/Pi ratio- did you mean to reverse this here at the end?
Author Response
Comment: Please revise your manuscript to correct all of the grammatical errors.
Response: We thank the reviewer for their critical comments. Appropriate changes are now made in the revised manuscript.
Comment: Line 65 – Should be Mg2+. All charges should be superscripts.
Response: We apologise for the mistake. Appropriate changes are now made in the revised manuscript.
Comment: Line 141 – “Table X”- what or where is Table X?
Response: We apologise for the mistake. Appropriate changes are now made in the revised manuscript.
Comment: Line 237 – “PO4” should be PO42-
Response: We apologise for the mistake. Appropriate changes are now made in the revised manuscript.
Comment: Line 246 - “promoted by bone sialoprotein (BSP), a calcium is a calcium-binding small integrin-binding ligand…”. Please rephrase this to clarify your meaning.
Response: We apologise for the mistake. The line is now rephrased in the revised manuscript.
Comment: Line 404- “TNAP in inevitable…”. I’m not sure what you are trying to say here but “inevitable” doesn’t seem to be the correct word?
Response: We apologise for the mistake. The phrase is now corrected in the revised manuscript.
Comment: Line 488 – “NNP1-/-“, should this be NPP1-/-?
Response: We apologise for the mistake. Appropriate changes are now made in the revised manuscript.
Comment: Line 553 – “Pi/PPi ratio”, you have previously referred to this in your manuscript as the PPi/Pi ratio- did you mean to reverse this here at the end?
Response: We apologise for the mistake. It is actually Pi/PPi and therefore appropriate changes are now made in the revised manuscript.